# Nationwide analysis of sex differences in waiting times for cataract surgery in Sweden between 2010 and 2022

Philip Jute[1] & Gustav Stålhammar [1,2] ✉

## Abstract

**Background** Sex-based disparities in healthcare access remain a global challenge. We aimed to investigate differences in waiting times for cataract surgery between males and females in Sweden, hypothesizing that such disparities might persist even within a universal healthcare system.

**Methods** We performed a nationwide retrospective cohort study using data from the Swedish National Cataract Register, which includes over 93% of all cataract surgeries. A total of 1,413,652 patients over 40 years of age who underwent cataract surgery between 2010 and 2022 were included. Exclusions were applied to those with waiting times exceeding 24 months and those residing outside Sweden. The primary outcome was waiting time between preoperative assessment and surgery, stratified by visual acuity, region, and demographic and clinical factors.

**Results** Here we show a mean waiting time of 64 days (standard deviation 126) for females and 60 days (standard deviation 102) for males ($P < 0.001$). This difference persists across all visual acuity strata and regions. A linear mixed-effects model with region as a random intercept indicates that males have a 3.3-day shorter waiting time compared to females ($P < 0.001$). Multivariate hazards regression identifies female sex, older age, specific comorbidities, and region of residence as significant predictors of longer waiting times. Although overall waiting times decrease over the study period, the sex-based gap remains consistent.

**Conclusions** We observe a persistent, albeit small, difference in waiting times favoring males. These findings highlight a systemic disparity that warrants further investigation and targeted interventions to ensure equitable access to cataract care.

## Plain language summary

This study looked at differences in waiting times for cataract surgery between males and females in Sweden, using data from over 1.4 million patients between 2010 and 2022. Cataracts cause clouding of the eye's lens and require surgery for treatment. Females waited an average of 64 days for surgery compared to 60 days for males. This small but consistent difference was seen across all levels of vision impairment and in all regions of Sweden. Even after accounting for other factors, including age, other eye conditions, and region, females still faced longer delays. While these differences are unlikely to affect health outcomes, they may point to inequities in the healthcare system. Efforts are needed to ensure fair and equal access to cataract surgery, regardless of sex.

Cataract remains the leading cause of blindness worldwide and represents an important public health challenge[1,2]. In Sweden, where healthcare is largely tax-financed, over 140,000 cataract surgeries are performed annually, representing approximately 1.4% of the country's 10-million population[3].

Previous research has identified female sex as a risk factor for cataract, even after adjusting for age and accounting for mortality as a competing risk[4,5]. Globally, a greater proportion of women than men experience blindness or visual impairment, and this disparity is projected to increase in the future[6]. Paradoxically, female sex has also been recognized as a barrier to accessing cataract surgery in Asia and Africa, and sex-based differences in ocular comorbidities, surgical complications, and preoperative best-

corrected visual acuity (BCVA) have been documented in American and European cohorts[7–11]. In Sweden, we recently demonstrated that BCVA at the time of surgery is comparable between sexes, suggesting that differences in cataract surgery rates are unlikely to reflect disparities in healthcare-seeking behavior or surgical admission criteria[12].

A previous study of 102,532 Swedish patients undergoing cataract surgery between 2010 and 2011 found that women experienced longer waiting times than men, even when stratified by similar levels of visual acuity[13]. Despite ongoing efforts to provide equitable healthcare access, persistent sex-based disparities have been reported in several medical fields[14,15].

[1]St. Erik Eye Hospital, Stockholm, Sweden. [2]Department of Clinical Neuroscience, Division of Eye and Vision, Karolinska Institutet, Stockholm, Sweden.
✉e-mail: gustav.stalhammar@ki.se

Here, we examine how sex-based differences in waiting times for cataract surgery in Sweden have evolved from 2010 to 2022. This study analyzes a cohort of over 1.4 million patients, covering approximately 93% of all cataract surgeries performed nationwide during this period, to provide a comprehensive understanding of trends in waiting times. We observe that females consistently wait longer for cataract surgery after admission, with an approximate difference of 3 days. This difference is not explained by variations in visual acuity, patient age, comorbidities, or region of residence.

## Methods

The study was approved by the Swedish Ethical Review Authority (reference 2022-00930-02) and adhered to the tenets of the Declaration of Helsinki. All relevant ethical regulations were followed. The requirement for informed consent was waived due to the study's retrospective nature, relying solely on previously collected data. This research did not involve any treatments, interventions, tests, analysis of biological samples, or collection of additional sensitive information. Additionally, we followed the Strengthening Reporting of Observational Studies in Epidemiology (STROBE) Guidelines.

### Inclusion and exclusion criteria

Data for this study were retrieved from the Swedish National Cataract Register (NCR), established in 1992 to document all cataract surgeries performed nationwide[3]. The register is governed by a steering committee comprising physicians representing both public and private healthcare sectors, academia, one nurse, and one patient representative. The NCR captures approximately 93% of all cataract surgeries conducted in Sweden, with data reliability continuously monitored and validated[16–18].

Patients eligible for inclusion were those over 40 years old undergoing a cataract operation between January 1, 2010, and December 31, 2022, following methods outlined in a previous study on cataract surgeries conducted in 2010 ($n = 1{,}482{,}725$)[13].

Patients aged 40 years or younger ($n = 66{,}495$) were excluded, as cataracts in this group are typically congenital, juvenile, or secondary to other diseases or trauma, meaning standard waiting time rules do not apply. There are no formal age-based cataract screening programs in Sweden (e.g., at 40 or 50 years), so using 40 years as a threshold does not inadvertently exclude patients who would have been screened otherwise. Furthermore, applying this age limit mirrors previous NCR studies examining sex differences and thus creates a comparable cohort[8,13].

Additionally, patients with waiting times over 24 months ($n = 687$) were excluded, as such extended delays are uncommon. These long waiting periods in the Swedish National Cataract Register (NCR) are likely due to registration errors or specific circumstances, such as a patient request for surgery by a particular surgeon. This was also an exclusion criterion in previous NCR studies, ensuring comparability[8,13].

Thirdly, 1816 patients residing outside Sweden were excluded, as clinicopathological data may be less reliable for these, and their waiting time for surgery may be influenced by factors non-typical to the standard situation in the Swedish healthcare system. Lastly, 75 patients without a recorded sex was excluded, leaving 1,413,652 patients for analysis (a flow diagram of the selection process is provided as Supplementary Fig. 1).

### Admission visit

In Sweden, the typical pathway for patients experiencing diminished visual acuity and other symptoms of cataracts often begins at a local optician. If cataracts are suspected, patients are referred to an ophthalmologist for further evaluation. Alternatively, patients may be referred by ophthalmologists who diagnose the condition during routine examinations for other eye-related issues. During the initial assessment, the patient's best-corrected visual acuity (BCVA) is measured by either an optometrist or an ophthalmic nurse. This is done using a KM-chart in a well-lit light box at a distance of three meters, where the BCVA is recorded on a decimal scale[19]. The test involves identifying the smallest line in which six out of seven letters are read correctly after subjective refraction. Patients may use their own spectacles if they prefer. The procedure for measuring BCVA has remained consistent throughout the study period. In addition, intraocular pressure is measured, and a detailed examination of the anterior segment, including the lens, is conducted using a slit-lamp biomicroscope. Biometry assessments, including keratometry and either optical or ultrasound biometry, are performed to calculate the precise power of the intraocular lens (IOL) to be implanted to achieve the desired refraction. Once the admission visit is completed, surgery is scheduled as soon as reasonably possible. Waiting times for the procedure can vary based on several factors, including the availability of surgical staff and operating rooms, patient travel constraints, personal preferences, and any coexisting conditions that might delay surgical intervention. In this study, the period from the admission visit to the day of surgery is defined as the waiting time. In some cases, patients are admitted based on examinations by other ophthalmologists outside the operating clinic. In such cases, the day the patient is admitted for surgery by the clinic performing the surgery is used as the admission visit.

The NCR form is completed after surgery, utilizing data from the admission visit examination and perioperative details, such as the use of capsule hooks, the type of antibiotic administered, and any complications that occurred. Additionally, the form records information about the performing clinic, the patient's personal identity number, cataract eye laterality, best-corrected visual acuity (BCVA), date of admission, specific health conditions (including pseudoexfoliation, cornea guttata, diabetes, macular disease, and glaucoma), type of surgery, and type of lens. The presence of specific health conditions is determined based on the examination during the admission visit as well as existing information in the patient's medical records, which may include free-text entries, International Classification of Diseases (ICD) codes, laboratory results, imaging studies, medication lists, and pathology reports. In Sweden, a person's sex is determined at birth by the attending healthcare staff and is encoded in the personal identity number, which reflects one of the two legal genders—male or female. No non-binary, unspecified, or third-gender options exist within this system.

### Statistics and reproducibility

Statistical significance was defined as a two-sided $P < 0.05$ unless otherwise specified. Continuous variables were assessed for normality using the Shapiro-Wilk test. If the data deviated from a normal distribution ($P < 0.05$), the Mann-Whitney U test was used for group comparisons; otherwise, Student's t-test was applied. Categorical baseline characteristics were compared using Pearson's chi-square test. To control for type I errors due to multiple comparisons, the two-stage step-up False Discovery Rate (FDR) method by Benjamini, Krieger, and Yekutieli was employed, with additional Bonferroni correction applied by multiplying P-values by the total number of statistical tests ($n = 38$). Yearly trends in waiting times were analyzed using linear regression models, including interaction terms to assess sex-based differences over time. A Kaplan-Meier survival curve for time to cataract surgery was generated, with differences assessed using the log-rank test. A multivariate Cox regression model was constructed to identify predictors of waiting times, with independent variables including sex, age, ocular comorbidities (pseudoexfoliation syndrome, cornea guttata, macular disease, diabetes, and glaucoma), and regional differences. To account for patients clustered within regions, we used a linear mixed-effects model including a random intercept for region and fixed effects for sex, age, BCVA, pseudoexfoliation, cornea guttata, diabetes, macular disease, and glaucoma. This multilevel approach accommodates regional baseline differences and refines patient-level effect estimates. Supplementary analyses included stratification by visual acuity groups, categorizing waiting times by decimal visual acuity equivalents. For each stratum, mean waiting times were compared between sexes using unpaired t-tests with Welch correction. All statistical analyses were conducted using IBM SPSS Statistics (version 29, Armonk, NY), GraphPad Prism (version 10.0.2, San Diego, CA, USA), and R (R Core Team, Vienna, Austria, 2022), with relevant packages including dplyr, lme4, lmerTest, ggplot2, tidyr, knitr, survminer, and survival.

## Table 1 | Baseline patient characteristics

| Variable | Females, $n$ = 828,515 | Males, $n$ = 585,137 | $P^a$ |
|---|---|---|---|
| Age, mean (SD) | 74.5 (8.6) | 74.3 (9.0) | <0.001 |
| BCVA operated eye[b], mean (SD) | | | <0.001 |
| Decimal scale | 0.46 (0.22) | 0.45 (0.23) | |
| LogMAR | 0.34 (0.21) | 0.35 (0.22) | |
| Snellen | 20/44 | 20/45 | |
| BCVA non-operated eye[c], mean (SD) | | | <0.001 |
| Decimal scale | 0.86 (0.34) | 0.89 (0.31) | |
| LogMAR | 0.07 (0.04) | 0.05 (0.04) | |
| Snellen | 20/23 | 20/23 | |
| Surgery type, $n$ (%) | | | <0.001 |
| Phaco. and IOL | 822,193 (99.24) | 579,420 (99.02) | |
| Phaco. and ACL | 532 (0.06) | 284 (0.05) | |
| Trab., phaco., and IOL | 273 (0.03) | 242 (0.04) | |
| Other[c] | 5517 (0.67) | 5191 (0.89) | |
| Lens material, $n$ (%) | | | <0.001 |
| Hydrophobic acrylic | 791,681 (95.55) | 559,208 (95.57) | |
| Hydrophilic acrylic | 30,920 (3.73) | 21,662 (3.70) | |
| Patient left aphakic | 1,670 (0.20) | 1,323 (0.23) | |
| Silicone | 1,335 (0.16) | 897 (0.15) | |
| PMMA | 219 (0.03) | 136 (0.02) | |
| Multifocal, material unspecified | 131 (0.02) | 125 (0.02) | |
| PMMA HSM | 26 (0.00) | 12 (0.00) | |
| Other | 2,520 (0.30) | 1,772 (0.30) | |
| Not specified | 13 (0.00) | 2 (0.00) | |
| Lens type, $n$ (%) | | | |
| Aspherical | 65,805 (7.94) | 43,203 (7.38) | <0.001 |
| Multifocal | 16,337 (1.97) | 13,670 (2.34) | <0.001 |
| Use of capsular tension rings, $n$ (%) | 15,207 (1.84) | 15,925 (2.07) | <0.001 |
| Postoperative endophthalmitis, $n$ (%) | 131 (0.02) | 154 (0.03) | <0.001 |
| Other systemic and ocular conditions, $n$ (%) | | | |
| Glaucoma, any type | 70,398 (8.50) | 53,113 (9.08) | <0.001 |
| Macular disease, any type | 129,126 (15.59) | 88,747 (15.17) | <0.001 |
| Pseudoexfoliations | 86,697 (10.46) | 47,472 (8.12) | <0.001 |
| Cornea Guttata | 25,294 (3.05) | 13,419 (2.29) | <0.001 |
| Diabetes, type I or II | 26,713 (3.23) | 32,663 (5.58) | <0.001 |

[a]Mann-Whitney $U$ test was used for continuous variables, and chi-square tests were used for categorical variables. Bonferroni correction was used to adjust $P$-values for multiple comparisons.
[b]Eye planned for surgery, or the first eye operated if both were treated in the same session. [c]Eye not planned for surgery, or the second eye operated if both were treated in the same session. [d]e.g., phacoemulsification (Phaco), intraocular lens (IOL), and simultaneous corneal surgery. *ACL* anterior chamber lens, *BCVA* best corrected visual acuity, *HSM* heparin surface modified, *IOL* intraocular lens, *PMMA* polymethyl methacrylate, *SD* standard deviation, Trab, trabeculectomy.

### Reporting summary

Further information on research design is available in the Nature Portfolio Reporting Summary linked to this article.

## Results

A total of 1,413,652 patients met the inclusion criteria, of whom 59% ($n$ = 828,515) were female. Compared with females, males had slightly better BCVA in the non-operated eye, were more likely to receive multifocal intraocular lenses (IOLs), more frequently required capsular tension rings, and experienced higher rates of postoperative endophthalmitis, whereas pseudoexfoliation was more common in females ($P < 0.001$ for all comparisons). Detailed baseline characteristics are presented in Table 1.

The average waiting time from preoperative assessment to surgery was 64 days for females (standard deviation [SD] 126) and 60 days for males (SD 102). A Shapiro-Wilk test confirmed non-normal distribution in both groups ($P < 0.001$ for both males and females); thus, a Mann-Whitney U test was conducted, indicating a significant difference in waiting-times between males and females ($W = 2.51 \times 10^{11}$, $P < 0.001$).

Differences in waiting times between females and males were observed across all visual acuity groups of the surgery eye, with females consistently experiencing longer average waiting times. These groups, categorized by visual acuity ($\leq 0.1$, 0.2, 0.3, 0.4, 0.5, 0.6, 0.7, 0.8, 0.9, and $\geq 1.0$ on the decimal scale, which is equivalent to $\leq 20/200$, 20/100, 20/66, 20/50, 20/40, 20/33, 20/28, 20/25, 20/22, and $\geq 20/20$ on the Snellen scale, and 1.0, 0.7, 0.52, 0.40, 0.30, 0.22, 0.15, 0.10, 0.05, and 0.0 on the LogMAR scale.), demonstrated statistically significant disparities. For instance, in the $\leq 0.1$ group, females had an average waiting time of 63 days (SD 71), compared to 57 days (SD 66 days) for males, a difference of 7 days (SE < 1 day, $P < 0.001$). Similar differences were evident across all other groups, with statistical significance consistently retained after adjusting for multiple comparisons using FDR. Unpaired $t$-tests with Welch correction confirmed statistically significant differences between males and females in all groups, with all $P$ values < 0.001. The magnitude of differences ranged from 2 days (SE 18, $P < 0.001$) in the BCVA 0.7 group, to 7 days (SE 18, $P < 0.001$) in the BCVA $\leq 0.1$ group (Fig. 1). These findings highlight a consistent trend of longer waiting times for females across all levels of preoperative visual acuity.

Two separate sensitivity analyses were performed to assess whether excluding certain subpopulations affected the primary findings. First, we reexamined the cohort of 66,495 patients who were $\leq 40$ years old; significant sex-based differences in waiting times persisted in this subgroup. Second, we evaluated the small subset of 687 patients with waiting times >24 months; in that group, the observed differences were explained by patient-specific factors rather than sex. Finally, re-including all excluded patients in the main analysis did not alter the overall conclusions, indicating that the results were robust (Supplementary Method: Sensitivity Analyses, and Supplementary Table 1).

Regional differences in waiting times were also evident, with the largest discrepancy observed in region 17 (mean difference: 9 days) and the smallest in region 13 (mean difference: 1 day, Fig. 2). Females had significantly longer waiting times in all regions (unpaired $t$-tests with Welch correction, $P < 0.001$ for all comparisons, adjusted for multiple comparisons using FDR). A full list of regional designations is available in Supplementary Table 2.

For regression analyses, regions were categorized into four tiers based on the magnitude of waiting time differences between genders, with Tier 1 representing regions with the smallest gender differences and Tier 4 those with the largest.

When treating waiting time for cataract surgery as a time-to-event analysis, females experienced longer cataract-surgery-free periods than males, indicating a longer delay in undergoing surgery. At 30 days after admission, the estimated Kaplan-Meier cataract-surgery-free survival was 64.9% (95% CI 64.8–65.0) for females and 62.2% (95% CI 62.1–62.4) for males. At 60 days, the survival was 39.7% (95% CI 39.6–39.9) for females and 37.1% (95% CI 36.9–37.2) for males. By 90 days, the survival was 23.4% (95% CI 23.4–23.5) for females and 21.3% (95% CI 21.2–21.4) for males, and at 120 days, it declined to 13.7% (95% CI 13.6–13.7) for females and 12.1% (95% CI 12.1–12.2) for males (Fig. 3A).

### Multilevel linear mixed-effects modeling

To account for clustering by region and to assess individual-level predictors of waiting time, we fit a linear mixed-effects model using region as a random intercept. The fixed effects included sex, age, best-corrected visual acuity (BCVA), and comorbidities (pseudoexfoliation, cornea guttata, diabetes,

**Fig. 1 | Comparison of mean waiting times for females (black) and males (green) across best-corrected visual acuity (BCVA) strata.** BCVA is reported on a decimal scale (e.g., Snellen 20/20 and LogMAR 0.00 correspond to 1.0, while Snellen 20/200 and LogMAR 1.00 correspond to 0.1). **a** Box plots of waiting time for cataract surgery by BCVA at the time of admission. The central horizontal line in each box indicates the median waiting time, whereas the dotted line represents the mean. Whiskers extend to the most extreme values within 1.5× the interquartile range (IQR). In all BCVA groups, females had significantly longer mean waiting times than males ($P < 0.001$, unpaired $t$-tests with Welch correction), q < Q after False Discovery Rate adjustment. **b** Volcano plot illustrating q values (on a $-\log_{10}$ scale) plotted against the magnitude of differences in mean waiting times between females and males. Points further from the origin indicate larger differences and greater statistical significance. **c** Line plot visualizing the mean waiting times for females and males across BCVA groups. Error bars represent one standard deviation. Number of independent patients: BCVA ≤ 0.1, $n = 39{,}683$; 0.2, $n = 114{,}512$; 0.3, $n = 268{,}351$; 0.4, $n = 86{,}668$; 0.5, $n = 203{,}373$; 0.6, $n = 442{,}630$; 0.7, $n = 47{,}694$; 0.8, $n = 102{,}807$; 0.9, $n = 67{,}439$; ≥1.0, $n = 40{,}496$.

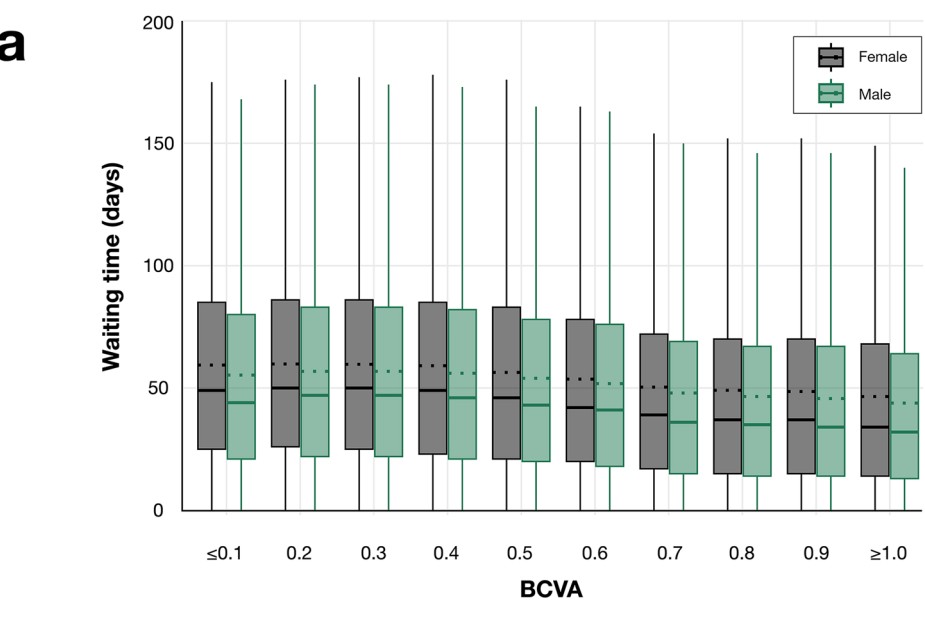

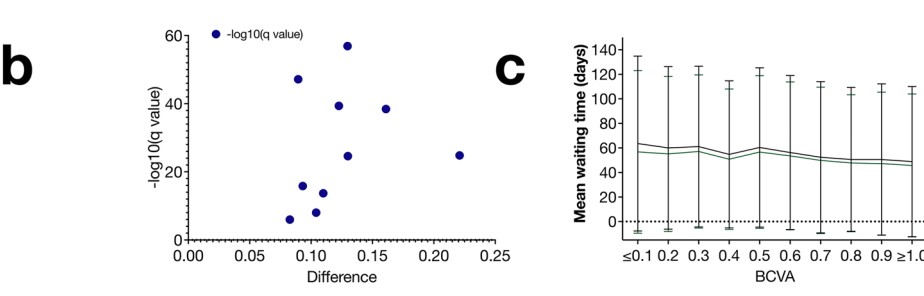

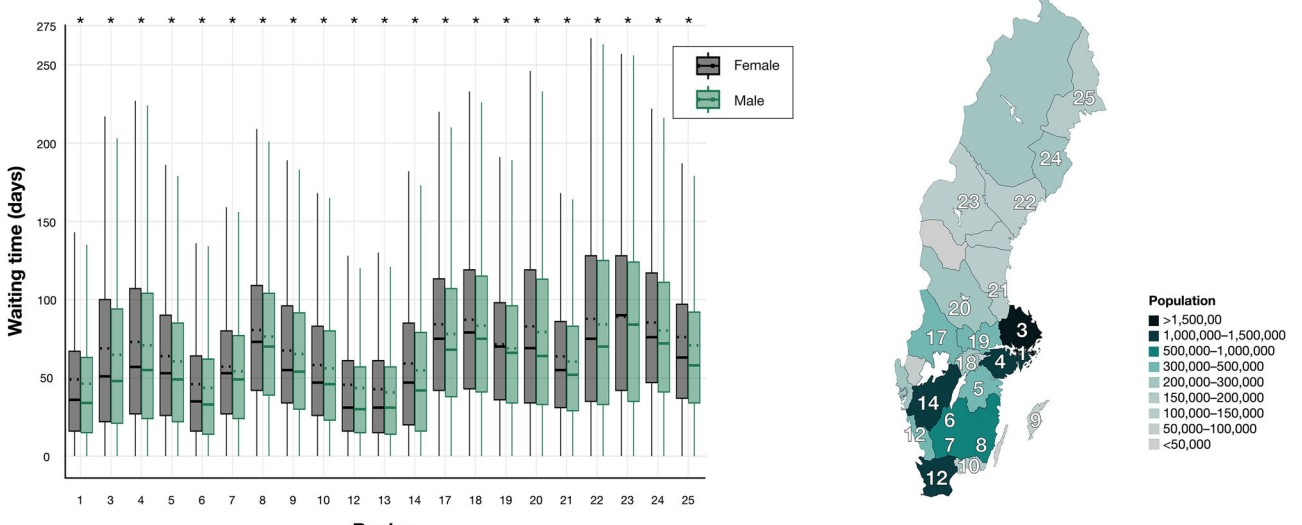

**Fig. 2 | Waiting time for females (black) and males (green) across Swedish regions.** Box plots display waiting times (in days) for cataract surgery stratified by gender across Swedish regions. The central line in each box indicates the median value and the dotted line denotes the mean. Whiskers extend to the most extreme observations within 1.5 times the interquartile range (IQR). The greatest discrepancy in waiting times was observed in region 17 (9 days), while the smallest was in region 13 (1 day). Asterisks denote statistically significant findings (q < Q after False Discovery Rate adjustment). The regions are highlighted on a map of Sweden, with shading representing the population in approximately corresponding areas. Number of independent patients: Region 1 $n = 289{,}596$; Region 3 $n = 53{,}341$; Region 4 $n = 47{,}801$; Region 5 $n = 56{,}030$; Region 6 $n = 51{,}562$; Region 7 $n = 26{,}832$; Region 8 $n = 36{,}115$; Region 9 $n = 9454$; Region 10 $n = 27{,}788$; Region 12 $n = 216{,}473$; Region 13 $n = 53{,}994$; Region 14 $n = 221{,}283$; Region 17 $n = 34{,}970$; Region 18 $n = 35{,}303$; Region 19 $n = 39{,}429$; Region 20 $n = 47{,}833$; Region 21 $n = 49{,}146$; Region 22 $n = 37{,}149$; Region 23 $n = 16{,}965$; Region 24 $n = 33{,}780$; Region 25 $n = 28{,}808$.

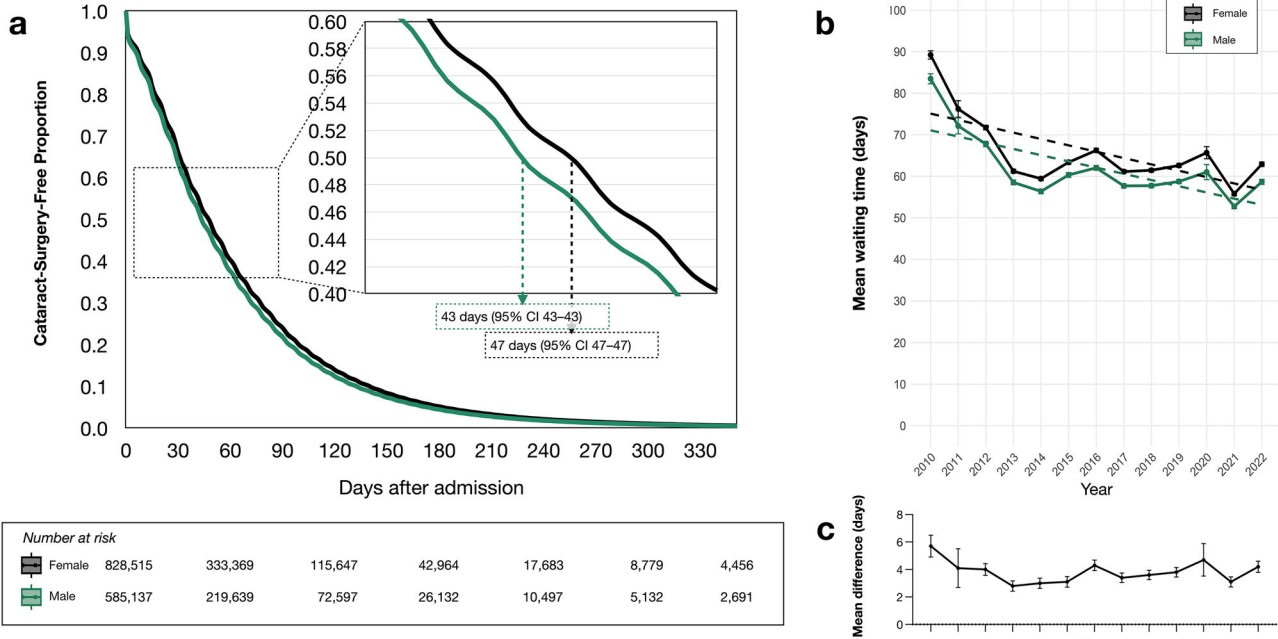

**Fig. 3 | Cataract-surgery-free survival and trends in cataract waiting times (2010–2022). a** Kaplan-Meier curve of cataract-surgery-free survival stratified by sex, showing a significant difference (Log-rank $P < 0.001$). The median cataract-surgery-free survival was 43 days for males (95% confidence interval [CI]: 43–43) and 47 days for females (95% CI: 47–47). **b** Annual mean waiting times from preoperative assessment to cataract surgery for females and males from 2010 to 2022. Error bars indicate 95% CIs. Overall waiting times decreased, following a slightly U-shaped trend over the study period. Dashed lines represent linear regression fits, which show a significant negative association between year and waiting time ($P < 0.001$; estimate: −1.38 days/year). The interaction term between year and sex was not significant (estimate: 0.063 days/year, $P = 0.16$), indicating similar trends in waiting times across sexes. **c** Annual mean difference in waiting times (in days) between females and males, calculated as the mean waiting time for females minus the mean waiting time for males. Error bars represent the standard deviation of the mean difference. Number of independent patients: Year 2010 $n = 80,822$; 2011 $n = 84,878$; 2012 $n = 87,036$; 2013 $n = 100,635$; 2014 $n = 103,362$; 2015 $n = 105,583$; 2016 $n = 111,951$; 2017 $n = 116,942$; 2018 $n = 121,427$; 2019 $n = 123,376$; 2020 $n = 107,961$; 2021 $n = 125,834$; 2022 $n = 143,845$.

macular disease, and glaucoma). Twenty-one Swedish regions were represented in the model. The fixed-effects analysis showed that males had significantly shorter waiting times than females ($\beta = -3.3$ days, $P < 0.001$). Older age was associated with longer waiting times ($\beta = 0.6$ days, $P < 0.001$), and better BCVA at admission corresponded to shorter delays ($\beta = -1.8$ days, $P < 0.001$). Pseudoexfoliation, cornea guttata, diabetes, macular disease, and glaucoma were also significantly linked to variations in waiting times (all $P < 0.001$ except macular disease, $P = 0.01$).

### Table 2 | Results of the Linear Mixed-Effects Model for Waiting Time

| Term | β (days) | S.E. (days) | t | P* |
|---|---|---|---|---|
| (Intercept) | 31.2 | 3.6 | 8.92 | <0.001 |
| Sex (male vs. female) | −3.3 | 0.09 | −33.03 | <0.001 |
| Age (years)ᵃ | 0.6 | 0.03 | 75.43 | <0.001 |
| BCVA | −1.8 | 0.03 | −7.59 | <0.001 |
| Pseudoexfoliation | 1.2 | 0.18 | 7.08 | <0.001 |
| Cornea Guttata | 4.8 | 0.33 | 14.65 | <0.001 |
| Diabetes | 9.6 | 0.24 | 36.87 | <0.001 |
| Macular Disease | 0.6 | 0.15 | 3.49 | 0.01 |
| Glaucoma | 3.3 | 0.18 | 16.69 | <0.001 |

*Bonferroni correction was applied to the *P* values to adjust for multiple comparisons. ᵃPatient age at the time of admission for cataract surgery. The model formula "Waiting time ~ Sex + Age + BCVA + Pseudoexfoliation + Cornea Guttata + Diabetes + Macular Disease + Glaucoma + (1 | Region)" denotes a linear mixed-effects model where *Waiting time* is the dependent variable (outcome). The terms before "(1 | Region)" are fixed effects, estimating how each variable (e.g., Sex, Age) relates to waiting time across all patients. The term "(1 | Region)" specifies a random intercept for each region, accounting for overall differences among regions beyond the fixed effects. *BCVA* best corrected visual acuity, measured per increasing step on the decimal scale (e.g., from 0.1 to 0.2, from 0.9 to 1.0, etc.). *S.E* standard error.

The random-effects analysis revealed that regional factors accounted for part of the variation in waiting times, with a random intercept variance of approximately 8.4 days (SD 15.9 days). The residual variance was about 119 days (SD 59.7 days), indicating that although patient-level variables explained a substantial proportion of the total variation, regional differences still contributed. Full results, including estimates for each fixed effect, confidence intervals, and random intercepts per region, are provided in Table 2.

### Multivariate Cox regression

Next, we performed a multivariate Cox regression analysis to investigate the effect of various factors on waiting time for cataract surgery (Table 3). The analyzed variables included sex (male vs. female), age, baseline visual acuity of the operated eye (BCVA), specific health conditions (presence of pseudoexfoliation, cornea guttata, diabetes, macular disease, and glaucoma), and region-specific waiting time tiers. All covariates, including patient sex, were found to be independent predictors of waiting time after applying Bonferroni adjustment to *P* values. Notably, diabetes (type I or II) emerged as a significant predictor of shorter waiting time (Hazard ratio: 0.88, 95% CI: 0.87–0.89).

### Time trends

An analysis of time trends in cataract surgery waiting times from 2010 to 2022 revealed a consistent decrease over the study period. In 2010, the average waiting time was approximately 89 days for females and 84 days for males, decreasing to around 63 days and 59 days, respectively, by 2022. A linear regression model—including an interaction term to assess gender differences—indicated a significant negative association between year and waiting time, with waiting times decreasing by an estimated 1.38 days per year ($P < 0.001$). The interaction term between year and sex was not statistically significant (Estimate = 0.063 days/year, $P = 0.16$), suggesting that

**Table 3 | Multivariate Cox Regression: Predictors of Time to Cataract Surgery**

| Variable | B | S.E. | Wald Test | P* | Exp(B) | 95% Confidence Interval |
|---|---|---|---|---|---|---|
| Sex (Male) | −0.064 | 0.003 | 608.7 | <0.001 | 0.938 | 0.934–0.943 |
| Age[a] | −0.007 | <0.001 | 2,708.2 | <0.001 | 0.993 | 0.992–0.993 |
| BCVA operated eye | 0.172 | 0.006 | 838.2 | <0.001 | 1.188 | 1.174–1.202 |
| Pseudoexfoliations | 0.077 | 0.005 | 257.5 | <0.001 | 1.080 | 1.070–1.091 |
| Cornea Guttata | −0.032 | 0.008 | 14.9 | <0.001 | 0.968 | 0.952–0.984 |
| Diabetes, type I or II | −0.127 | 0.006 | 407.0 | <0.001 | 0.881 | 0.870–0.892 |
| Macular Disease, any type | 0.02 | 0.004 | 29.5 | <0.001 | 1.020 | 1.013–1.027 |
| Glaucoma, any type | −0.052 | 0.005 | 130.2 | <0.001 | 0.949 | 0.941–0.958 |
| Region tier[b] | −0.125 | 0.001 | 10,144.3 | <0.001 | 0.883 | 0.881–0.885 |

*Bonferroni correction was applied to the P values to adjust for multiple comparisons. [a]Patient age at the time of admission for cataract surgery. [b]Regions were classified into four tiers based on average waiting time for cataract surgery, with Tier 1 containing regions with the longest waiting time, and Tier 4 those with the shortest. BCVA best corrected visual acuity, measured per increasing step on the decimal scale (e.g., from 0.1 to 0.2, from 0.9 to 1.0, etc.). S.E standard error.

the rate of decline was similar for both genders. Thus, although overall waiting times for cataract surgery decreased, the relative difference between males and females remained stable (Figs. 3B, 3C).

## Discussion

This study confirms persistent, albeit small, sex-based differences in waiting times for cataract surgery in Sweden over a 13-year period, with female patients consistently experiencing longer delays than their male counterparts. Despite efforts to improve healthcare equity, this disparity has remained stable and statistically significant across the study period. Importantly, these differences are unlikely to be explained by clinical factors, such as visual acuity at the time of admission for surgery, age, or comorbidities, which were comparable between sexes or adjusted for in the analyses.

Cataract is typically a slowly progressive condition, and although the absolute differences of a few days in waiting times are small and unlikely to have a major impact on health or long-term well-being, they may reflect underlying systemic inequalities or biases that warrant further investigation. In 2004, the National Board of Health and Welfare (NBHW) identified differences in access to care for females and elderly patients as examples of discrimination[20]. The persistent disparity observed suggests the need for targeted interventions to achieve more equitable access to surgery.

Our findings align with earlier research indicating sex-based disparities in waiting times for cataract surgery. A 2010–2011 study of Swedish patients reported similar patterns of longer waiting times for women[13]. Smirthwaite and colleagues based their conclusions on focus interviews with ophthalmologists, noting that certain traits, such as assertiveness and patterns of seeking care, were ascribed differently to male and female patients[21]. While the 2010–2011 study uses the term "gender" when making these comparisons, the Swedish National Cataract Register (NCR) only contains information on legal sex—that is, female or male—determined by each patient's personal identity number. Sex typically refers to a person's biological characteristics, whereas gender refers to socially constructed roles and norms[22]. Therefore, although "gender" is used in these studies, the actual data analyzed reflect only legal sex. In Sweden, there are two legally recognized sexes (female and male), and even individuals with intersex conditions are registered as one of these two categories. As a result, the previous studies effectively compare "sex" differences rather than "gender" differences, using the same inclusion criteria and dataset structure as the present study.

Sex-based disparities are not unique to Sweden; studies from other regions have reported similar differences in access to ophthalmic care, with women facing barriers to timely treatment even in high-resource settings[10]. Such disparities are often attributed to social, economic, or systemic factors rather than biological differences in disease burden or progression[23,24].

In addition to sex, other predictors of waiting time included age, home region, and specific comorbidities. Older patients tended to wait longer,

potentially reflecting the prioritization of working-age individuals or the need to address comorbidities before surgery. Regional variations were notable, with differences likely driven by healthcare resource allocation, availability of surgeons, and logistical factors such as travel distance. Previous studies have also highlighted variations in complication rates of cataract surgery at regional, clinic, and even individual surgeon levels[25].

Although differences in surgical technique and complications between sexes were observed, these were minor and unlikely to account for the disparity in waiting times. For example, capsular tension rings and postoperative endophthalmitis were slightly more common in male patients, while pseudoexfoliation syndrome was more frequent among females. While statistically significant due to the large sample size, these differences are of limited clinical relevance.

The period from 2020 to 2022 encompassed the global COVID-19 pandemic, which significantly disrupted healthcare systems worldwide, including elective surgical procedures such as cataract surgery. During the initial waves of the pandemic, many healthcare resources were reallocated to manage COVID-19 cases, leading to delays in non-emergency surgeries. Our analysis revealed a generally decreasing trend in waiting times, with a slightly U-shaped pattern and notable fluctuations from 2020 to 2022 that are likely attributable to pandemic-related disruptions. Specifically, the increase in waiting times during the peak pandemic years may reflect reduced surgical capacity, staff shortages, and heightened safety protocols to prevent virus transmission. Although our multivariate analyses accounted for temporal trends, the nuanced impact of the pandemic on sex-based disparities remains underexplored. Future studies should investigate whether the pandemic exacerbated existing inequities or introduced new barriers to timely cataract surgery access for different sexes.

Regional disparities in waiting times for cataract surgery were prominent in our findings. In the context of Sweden's tax-funded healthcare system, resource availability in terms of funding and surgical facilities is relatively uniform across regions. However, a significant variation exists in the availability of cataract surgeons. Some regions experience challenges in attracting and retaining ophthalmologists, which can lead to longer waiting times for surgeries in those areas. Additionally, in some regions, travel time to facilities that offer cataract surgery may be very long. This uneven distribution of cataract surgeons and geographical accessibility may be a driver of regional differences in waiting times.

It is unclear whether the shortage of cataract surgeons in certain regions directly influences the difference in waiting times between males and females. Potential factors contributing to this gender-specific disparity may include differential referral patterns, varying patient preferences, or implicit biases in clinical decision-making. Further investigation is needed to elucidate the exact mechanisms behind this disparity. Understanding why female patients are more affected by regional surgeon availability could inform targeted strategies to mitigate these inequities and ensure more equitable access to cataract surgery across all regions.

A major strength of this study is its comprehensive dataset, encompassing over 1.4 million patients and covering approximately 93% of all cataract surgeries performed in Sweden during the study period. This extensive coverage ensures a robust representation of real-world clinical practice and reduces the risk of selection bias.

However, the study's retrospective registry-based design limits causal inferences. While we identified significant predictors of waiting time, the underlying reasons for sex-based disparities remain unclear. Additionally, the accuracy of the findings relies on the quality of data entered into the Swedish National Cataract Register. Although previous audits have validated the register's reliability, occasional inaccuracies in reporting cannot be excluded[16].

This study has several other limitations that warrant consideration. Firstly, the Swedish National Cataract Register does not capture data on patients' race or ethnicity. As a result, we were unable to assess whether racial or ethnic disparities exist in waiting times for cataract surgery, which could be an important factor in understanding broader healthcare inequities. Secondly, while we utilized regional classifications as proxies for geographic factors, we did not directly measure individual patients' driving distances or travel times to surgical centers. Future research should aim to incorporate more granular geographic data, such as precise patient locations and transportation logistics, to better evaluate the impact of distance on waiting times. Additionally, exploring other potential confounders, such as socioeconomic status and insurance coverage, could provide a more comprehensive understanding of the factors contributing to sex-based and regional disparities in cataract surgery waiting times.

The persistence of sex-based differences in waiting times for cataract surgery highlights the need for targeted interventions to address healthcare inequities. Potential strategies include enhancing referral practices, standardizing prioritization criteria, and ensuring adequate staffing across regions. Further research is needed to elucidate the root causes of these disparities, including qualitative studies to explore potential biases in clinical decision-making and patient preferences.

Finally, while the observed differences may not have a significant impact on clinical outcomes, they could contribute to perceptions of inequity and undermine trust in the healthcare system. Addressing even small disparities is essential for ensuring that healthcare delivery is both equitable and perceived as fair by all patients.

In conclusion, females in Sweden experienced slightly longer waiting times for cataract surgery than males during the period 2010–2022. This disparity was consistent across the study period and remained significant after adjusting for clinical and demographic factors. While the differences are unlikely to have major clinical implications, they highlight the need for continued efforts to achieve equitable access to ophthalmic care.

## Data availability
Source data for the figures are available as Supplementary Data. Patient-level data analyzed in this study are available from the Swedish National Cataract Register (https://rcsyd.se/anslutna-register/nationella-kataraktregistret). Access to these data requires approval from the Swedish Ethical Review Authority and the Swedish National Cataract Register's record keeper. Requests for data can be submitted through the register's website and must comply with Swedish regulations governing the use of healthcare data for research purposes.

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

## Acknowledgements

Support for this study was provided to Gustav Stålhammar from: Region Stockholm (RS-2019-1138). The funding organization had no role in the design or conduct of this study.

## Author contributions

Philip Jute: Conceptualization, Writing – Review & Editing. Gustav Stålhammar: Conceptualization, Methodology, Software, Formal Analysis, Investigation, Data curation, Writing – Original Draft, Visualization, Project administration, Funding acquisition.

## Funding

## Competing interests

The authors declare no competing interests. Gustav Stålhammar was previously an Editorial Board Member for Communications Medicine, but was not involved in the editorial review or peer review, nor in the decision to publish this article.
