## [Transparent Peer Review file · Communications Medicine]

Nationwide analysis of sex differences in waiting times for cataract surgery in Sweden between 2010 and 2022

Corresponding Author: Dr Gustav Stålhammar

Version 0:

Reviewer comments:

Reviewer #1

(Remarks to the Author)

The authors present a well-designed, longitudinal study of sex differences in wait times for cataract surgery in Sweden. The statistical analyses are well described and easy to follow, especially for the purpose of reproducibility. The major claim of the paper has been previously described in a Swedish population as mentioned by the authors (Smirthwaite et al.). However, the longitudinal analysis including 10 years' worth of data, geographical depictions, and cox regression analyses strengthen previous claims.

Regarding obtaining of clinical characteristics, I would suggest minor clarification. How were the variables (glaucoma, macular disease, PXE, guttae, and diabetes) reported within the NCR database? For example, was the presence of corneal guttae from clinical exam information or diagnostic information (e.g. ICD codes)?

Reviewer #2

(Remarks to the Author)

Line 58 :change from like to including

Line 60: Update to: Efforts are needed to ensure fair and equal access to cataract surgery, regardless of sex.

Methods:

- Can the authors detail their reasoning for excluding 40-year-olds as cataract screening usually starts at this age.
- Were any sensitivity analyses conducted to see if any of the variables assessed in the study applied to why the 687 patients were delayed, rather than hypothesis of "These long waiting periods in the Swedish National Cataract Register (NCR) 98 are likely due to registration errors or specific circumstances, such as a patient request for surgery by a particular surgeon."
- Line 104-109 move up to first paragraph of methods.
- Were there any intersex patients? More agencies are highlighting the inclusion within research studies. Were individuals that are intersex excluded?
- Why was ethnicity not considered in this study? Other research has demonstrated that ethnic/racial differences related to cataract surgery occur.
- No mention of multilevel modeling, patients are clustered within regions and patient level data is being used. There is a nested structure (hierarchical modeling) needed in the modeling. This will provide a true effect of the region.

Results

- Please provide p-values for findings in paragraph 1
- Line 200-203 seems to be a point for the discussion section and further discussion there.
- When an acronym is used in a table/figure, please provide a footnote for what it means (ex: BCVA = Best Corrective Visual Acuity)
- Please provide a supplemental figure of a STROBE flow chart

Discussion

- Further discussion of the impact of the pandemic on findings are needed.
- Contextualization with other studies needs additional details. The current study is examining sex, but comparison study is assessing gender (women). These are separate – thus there needs to either be same comparison or at least mention that sex and gender do not necessarily perfectly correlate. Or please double check literature, and insure they state woman rather

than female.

- Additional context on region, as I see in the map population size is mentioned in the map, but details around this is not discussed. Is region more impactful due to population size or driving distance, resources within the region?
- Additional limitations: Race/ethnicity not considered, distance/driving – region was utilized as a proxy but addition discussion as a limitation/future research needed to assess distance is important.

Version 1:

Reviewer comments:

Reviewer #1

(Remarks to the Author)

The authors have adequately addressed my previous review comments.

Reviewer #2

(Remarks to the Author)

I thank the authors for addressing my edits and comments. I have no additional edits or comments.

POINT-BY-POINT RESPONSE FORM

Manuscript #: COMMSMED-24-1728

Manuscript title: Sex Differences in Waiting Times for Cataract Surgery in Sweden, 2010–2022: Nationwide Analysis of 1.4 Million Patients

Reviewer #1	Author's Response	Change in (new version of) the Manuscript
The authors present a well-designed, longitudinal study of sex differences in wait times for cataract surgery in Sweden. The statistical analyses are well described and easy to follow, especially for the purpose of reproducibility. The major claim of the paper has been previously described in a Swedish population as mentioned by the authors (Smirthwaite et al.). However, the longitudinal analysis including 10 years' worth of data, geographical depictions, and cox regression analyses strengthen previous claims.	Thank you!	-
Regarding obtaining of clinical characteristics, I would suggest minor clarification. How were the variables (glaucoma, macular disease, PXE, guttae, and diabetes) reported within the NCR database? For example, was the presence of corneal guttae from clinical exam information or diagnostic information (e.g. ICD codes)?	The presence of specific health conditions is determined based on the examination during the admission visit as well as existing information in the patient's medical records, which may include free-text entries, International Classification of Diseases (ICD) codes, laboratory results, imaging studies, medication lists, and pathology reports. This information has now been added to the "Admission visit" section	Admission visit, row 142–157
Reviewer #2	Author's Response	Change in (new version of) the Manuscript
Line 58 :change from like to including	Thank you for your helpful comments. Changed accordingly.	Plain language summary, row 62
Line 60: Update to: Efforts are needed to ensure fair and equal access to cataract surgery, regardless of sex.	Changed accordingly.	Plain language summary, row 64-65
Methods: • Can the authors detail their reasoning for excluding 40-year-olds	Patients aged 40 years or younger were excluded, as cataracts in this group are typically congenital, juvenile, or secondary to other diseases or trauma,	Inclusion and exclusion criteria, rows 109–112

as cataract screening usually starts at this age.	meaning standard waiting time rules do not apply. There are no formal age-based cataract screening programs in Sweden (e.g., at 40 or 50 years), so using 40 years as a threshold does not inadvertently exclude patients who would have been screened otherwise. Furthermore, applying this age limit mirrors previous NCR studies examining sex differences and thus creates a comparable cohort. This has been added to our description of inclusion and exclusion criteria	
Were any sensitivity analyses conducted to see if any of the variables assessed in the study applied to why the 687 patients were delayed, rather than hypothesis of "These long waiting periods in the Swedish National Cataract Register (NCR) 98 are likely due to registration errors or specific circumstances, such as a patient request for surgery by a particular surgeon."	We have now conducted sensitivity analyses to address the reviewer's comment. Specifically, we analyzed the 687 patients excluded due to waiting times exceeding 24 months, as well as the 66,495 patients who were ≤ 40 years old. While these analyses do not provide definitive explanations for the exceptionally long waiting times in some cases, they confirm that significant sex-based differences in waiting times persist within the excluded group of 66,495 patients aged ≤ 40 years. Conversely, in the smaller group of 687 patients with waiting times > 24 months, the differences appear to be influenced by specific patient factors rather than sex. These sensitivity analyses are now detailed in the supplementary material, including a supplementary table that presents the results of a multivariate Cox regression for the excluded patients.	Sensitivity analyses in supplementary material Supplementary table 1 Results, rows 209–215
Line 104-109 move up to first paragraph of methods.	Moved accordingly	Methods, row 89–95
Were there any intersex patients? More agencies are highlighting the inclusion within research studies. Were individuals that are intersex excluded?	In Sweden, there are only two legally recognized genders, male and female. In this study, sex was determined based on patients' personal identity numbers, which include the date of birth (year-month-day) followed by four digits. The third digit of this sequence designates sex, with even numbers assigned to females and odd numbers assigned to males (e.g., 19650210-8220 for a female born on February 10, 1965). As a result, the registry only allows for classification into these two sexes, meaning that	-

	individuals who are intersex would still be registered as either male or female.	
Why was ethnicity not considered in this study? Other research has demonstrated that ethnic/racial differences related to cataract surgery occur.	Patients' ethnicity is not recorded in the Swedish National Cataract Register and was therefore unavailable for this study. In Sweden, collecting and registering information on ethnicity or race is considered sensitive, and such data are generally not defined or systematically recorded in healthcare registers.	-
No mention of multilevel modeling, patients are clustered within regions and patient level data is being used. There is a nested structure (hierarchical modeling) needed in the modeling. This will provide a true effect of the region.	We agree that patients are nested within regions, creating a hierarchical data structure that necessitates a multilevel (hierarchical) modeling approach. To address this, we employed a linear mixed-effects model, specifying random intercepts for each region, thereby capturing the correlation among patients treated within the same region. In the multilevel (mixed-effects) model, males had shorter waiting times, older age and specific comorbidities were associated with longer waits, and a meaningful portion of the overall variance was explained by differences among regions.	Results, new "Multilevel Linear Mixed-Effects Modeling section", rows 233–248
Results Please provide p-values for findings in paragraph 1	P value added	Results, row 188
Line 200-203 seems to be a point for the discussion section and further discussion there.	We agree, these sentences have been moved to the discussion	Results, rows 318–328
When an acronym is used in a table/figure, please provide a footnote for what it means (ex: BCVA = Best Corrective Visual Acuity)	Abbreviations are spelled out	All tables and figure legends
Please provide a supplemental figure of a STROBE flow chart	A STROBE flow chart has now been added as Supplementary Figure 1 .	Supplementary material
Discussion Further discussion of the impact of the pandemic on findings are needed.	A paragraph has been added to the "Factors influencing waiting times" section in the Discussion, specifically discussing the COVID pandemic	Discussion, rows 318–328
Contextualization with other studies needs additional details. The current study is examining sex, but comparison study is assessing gender (women). These are separate – thus there needs to either be same comparison or at least mention that	We agree. Our findings align with earlier research indicating sex-based disparities in waiting times for cataract surgery. A 2010–2011 study of Swedish patients reported similar patterns of longer waiting times for women. ³³ Smirthwaite and colleagues based their conclusions	Discussion, rows 287–304

sex and gender do not necessarily perfectly correlate. Or please double check literature, and insure they state woman rather than female.	on focus interviews with ophthalmologists, noting that certain traits, such as assertiveness and patterns of seeking care, were ascribed differently to male and female patients.²¹ While the 2010–2011 study uses the term “gender” when making these comparisons, the Swedish National Cataract Register (NCR) only contains information on legal sex—that is, female or male—determined by each patient’s personal identity number. Sex typically refers to a person’s biological characteristics, whereas gender refers to socially constructed roles and norms.²² Therefore, although “gender” is used in these studies, the actual data analyzed reflect only legal sex. In Sweden, there are two legally recognized sexes (female and male), and even individuals with intersex conditions are registered as one of these two categories. As a result, the previous studies effectively compare “sex” differences rather than “gender” differences, using the same inclusion criteria and dataset structure as the present study. This information has been added to the discussion.	
Additional context on region, as I see in the map population size is mentioned in the map, but details around this is not discussed. Is region more impactful due to population size or driving distance, resources within the region?	A new section has been added to the Discussion: “Regional variations in waiting times” in which we discuss the regional variations, importance of the Swedish tax-funded health care system for distribution of resources, availability of ophthalmologists in different areas etc.	Discussion, “Regional Variations in Waiting Times” section, rows 330–345
Additional limitations: Race/ethnicity not considered, distance/driving – region was utilized as a proxy but addition discussion as a limitation/future research needed to assess distance is important.	These limitations have been added to the Limitations section	Discussion, Strengths and limitations, rows 357–367